# Generating Packed Rectilinear Display Text Layouts with Weighted Word Emphasis

Cheryl Lao*
University of Waterloo

Craig S. Kaplan†
University of Waterloo

Daniel Vogel‡
University of Waterloo

Jose Echevarria§
Adobe Research

Paul Asente¶
Adobe Research

## ABSTRACT

A common text layout style is a "packed rectilinear layout", in which non-overlapping word bounding boxes are packed so that their union forms a rectangle with no holes other than word and line spacing. Designing variations of these layouts while preserving word emphasis is a difficult and time-consuming process. We present a display text layout algorithm in which designers specify parameters that control the visual emphasis of words in these layouts. The number of possible layouts for a phrase follows the sequence of Big Schröder numbers as our algorithm involves the recursive subdivision of a rectangular bounding box. We conducted semi-structured interviews with graphic design experts to better understand their design decisions in creative typesetting. They rated the best-fitting layouts generated by our system to be very similar to designs that they would have created themselves.

**Index Terms:** Human-centered computing—Human computer interaction (HCI)—Interactive systems and tools—; Computer graphics—Graphics systems and interfaces——

## 1 INTRODUCTION

Display text layouts are stylized typographical arrangements consisting of short phrases, used for applications like headlines, advertisements, and logos. They require skill to design because they combine both typography and graphic design. This is in contrast with body text, which is relatively simple and uniform to lay out. Designers often need to emphasize certain words in a layout to convey the intended meaning of the phrase. However, the shapes and sizes of words has a direct effect on the layout and small changes to the text can have cascading effects on the overall layout, changing the emphasis. For example, Figure 1 is a layout generated using an Adobe Magic Text [1] template in which a small change to the text changes the emphasis of the layout from "healthy" to "how to." Designing aesthetically pleasing layouts that emphasize certain words is a common but time-consuming process because of the many possible layout variations for any given phrase.

The relative emphasis of words is a key factor in readability and semantics of the original phrase. Designers often wish to emphasize certain words in a layout, but they are also constrained by the shape of the layout, reading order, or the locations of less salient words. It is difficult to strike a balance between readability, semantics, and aesthetics. Our goal is to support designers in this task by generating variations of display text layouts that satisfy these constraints.

An automated and assisted display text system should ideally allow a user to specify parameters to control the visual emphasis of words in a layout without sacrificing its aesthetic quality. Existing

---

*e-mail: cheryl.lao@uwaterloo.ca

†e-mail: csk@uwaterloo.ca

‡e-mail: dvogel@uwaterloo.ca

§e-mail: echevarr@adobe.com

¶e-mail: research@asente.com

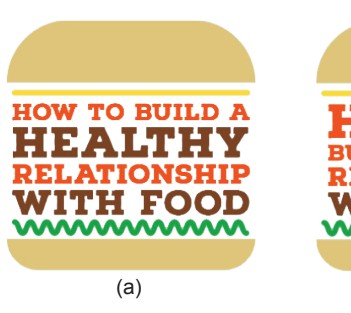
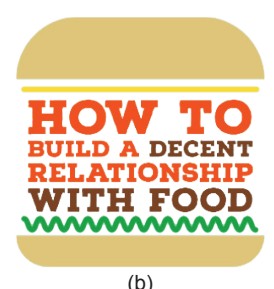

(a)                (b)

Figure 1: Example of unintended emphasis in display text with the template approach in Adobe Magic Text: (a) template layout and colour emphasize semantically important words; (b) changing the word 'HEALTHY' to 'DECENT' significantly alters the visual emphasis of words reducing the readability and saliency of the design. Note the new word is only one character shorter and colours of all words are unchanged; the difference is entirely due to the layout.

techniques for automatically generating 2D layouts from a given set of visual elements are mostly focused on different use cases, like magazines [11], photo collages [8], and other single-page graphic designs [16], which are less rigid in the relative placement of text elements than display text layouts.

In this work, we focus on packed rectilinear layouts, such as the example in Figure 1. These consist of words with non-overlapping bounding boxes packed so that the union of all bounding boxes forms a rectangle with no holes other than word and line spacing. While Adobe Magic Text [1] creates these layouts with the goal of achieving a specific aspect ratio, our goal is to achieve a specific level of relative emphasis between words in the layout. Our algorithm generates all possible packed rectilinear layouts for a phrase and prioritizes the layout variations based on their adherence to the desired relative emphasis of words. We also present the results from a set of semi-structured interviews with graphic design experts that aimed to build our understanding of design decisions in creative typesetting.

## 2 BACKGROUND

The automatic layout of visual elements has been an area of extensive research and implementation in commercial products, but typography imposes unique constraints on the visual layout process. Words need to be presented in an order consistent with the reading direction of the chosen script, and word emphasis depends on many visual factors.

**Automatic Layout Techniques** Hurst et al. [9] describe document layouts as *constrained optimisation problems* and differentiated between *micro-typography*, the low-level composition of text, and *macro-typography*, the overall appearance of the layout. Our work touches on both micro-typography in the alignment of words and macro-typography in overall layout decisions.

Existing work on graphic design can inform our development of a display text layout technique. Magazine covers share many

similarities with display text layouts, including the need for emphasis on certain elements and constraints on design proportions. A magazine cover layout typically consists of a large background image with blocks of text around the edges of the page. Existing machine-learning approaches for magazines take the salience of the background image into account, but they do not focus on the relative positioning among elements [11, 21, 23].

Grid-based layout algorithms divide a canvas into different areas [7, 10]. This approach works well for the layout of documents where margins between elements and irregular packing are permissible, but it cannot be easily extended to display text layouts where the relative placement of elements is also constrained by reading order.

Blocked Recursive Image Composition (BRIC) [4] is another visual layout technique that automates the creation of packing variations. This technique arranges a set of visual elements relative to one another spatially with constraints driven by recursive decomposition of the elements. BRIC respects element aspect ratios and includes precise spacing between elements unless adjustments are necessary to preserve the aspect ratio. Elements are represented in a binary tree where each internal node describes the alignment of its children.

Layout approaches that involve the relative placement of elements can be applied to display text layouts if they offer controls for the overall shape of the layout. Kraus [13] proposed a technique for tree-based automatic text layout. His method describes the relationships between words through alignment operators at internal nodes, where each operator describes the relative alignment of the node's children. These trees can be traversed to create layouts using the relative positioning between nodes. Other layout techniques, such as the one proposed by Piccoli et al., attempt to achieve similar word size for all words in a phrase [17].

Graphic Design Principles   The composition of text can be approached with layout principles that are widely used in graphic design. Bauerly et al. [5] presented two experiments that explored the effect of symmetry, balance, and quantity of construction elements on interface aesthetic judgments. In our work, we extend these principles and formalize the templates presented by Bauerly et al. in order to automatically generate visually pleasing layouts.

O'Donovan et al. [16] proposed an energy-based approach derived from design principles to analyze, create, and evaluate the design quality of layouts. In the evaluation stage, the importance of each element, labels specifying element alignment, and a grid-based segmentation are derived for an input layout. These are used as inputs to an energy function. The energy function also considers the visual salience of the image on the location of the text. Although their system produced visually pleasing results, the technique is very time-intensive and not interactive.

DesignScape [15] is another tool that provides layout suggestions for designers by varying attributes such as alignment and scale for design elements. Their tool provides layout options that can be selected as well as an adaptive interface that adjusts elements automatically with any change in the layout from the user.

Text Attributes   Legibility at-a-glance is a crucial feature of successful display text layouts. Sawyer et al. [19] explored which attributes make layouts legible upon a quick glance and compared these attributes across eight popular sans serif fonts. "Personality" is a concept that is used by designers to determine the font selection for different designs, but not a well-defined term. Researchers have tried to find empirical measures that are associated with certain moods using a subset of letters to determine font personality [14] and through crowd-sourced opinions on font connotations [20].

While past work has presented techniques for flexible layouts of visual elements in general, display text layouts have specific constraints, such as reading order and aspect ratio, and we focus this work on them. We present a technique for generating all possible packed rectilinear layouts for a text phrase and rank the layouts

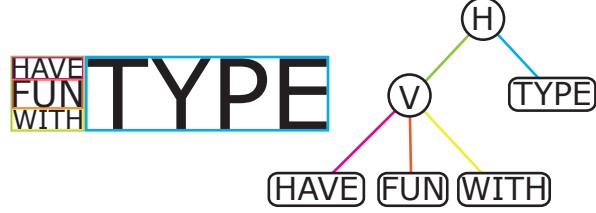

Figure 2: A layout with its corresponding tree structure. Here, H represents a horizontal alignment and V represents a vertical alignment between subtrees.

based on designer preferences and design principles. The design of our tool was guided by a series of interviews with expert designers.

## 3   TECHNIQUE

We present an algorithm for generating and ranking all packed rectangular layouts that are possible for a given phrase. The generated layouts adhere to these characteristics:

- each word must be to the right of or below the previous word in the phrase;

- the convex hull of all the words in the layout must closely approximate a rectangle;

- the layout must be filled with words, word spacing, or leading (the vertical space between lines).

Let $\vec{w} = (w_1 \ldots, w_n)$ be a sequence of $n$ words representing the phrase to be laid out, and let $\vec{e} = (e_1 \ldots, e_n)$ be a vector representing the designer's intended emphasis goal for each word, which we also refer to as the *emphasis schema*. For example, $\vec{e} = (4, 1, 1, 3)$ means the first word should be emphasized most, followed by the fourth word, with the remaining two words equally least emphasized. The numeric value of a characteristic, such as height or width of each word, can be represented using a characteristic vector $\vec{c} = (c_1 \ldots, c_n)$. These characteristics can be any parameterized attribute that contributes to word emphasis. Our goal is to compute an emphasis adherence score $E$ for every possible packed rectilinear layout for a given phrase.

We chose exhaustive generation of layouts because it allows us to find the optimal layouts that fit an emphasis schema and gives designers the maximum number of possible layouts to use as a template. This also provides a wider variety of layouts for us to use as examples when answering questions about aesthetics and design. Layouts that do not match the emphasis schema closely might also be valuable to the designer as a starting point for designs, so it is useful to present all variations as possibilities.

### 3.0.1   Algorithm Summary

The main steps for our layout algorithm are described below, with details in the following sections.

1. **Tree Generation:** Recursively generate all possible subdivisions of the input phrase. Each subdivision is a subtree where leaf nodes represent individual words and internal nodes represent subsets of the words in the phrase. The result is a list of all possible binary and non-binary trees with $n$ leaf nodes

2. **Layout Generation:** For each resulting tree structure, create two layouts by starting at the root and labelling each level as a horizontal or vertical alignment level, alternating between the two. Then, start at the leaf nodes and align horizontally by placing words adjacent to one another and scaling to the same

$\vec{e}$ = (1, 1, 1, 1)

$\vec{e}$ = (1, 2, 3, 4)

$\vec{e}$ = (4, 1, 1, 3)

$\vec{e}$ = (3, 3, 2, 1, 1, 4)

$\vec{e}$ = (1, 1, 1, 1, 2, 1, 1.2, 1.2)

$\vec{e}$ = (1, 1, 1, 1, 2, 1, 1.2, 1.2)

Figure 3: Each row shows the top 5 layouts for different emphasis goal vectors (highest to lowest match, left to right). Note that it is sometime impossible to create layouts that perfectly match the emphasis schema while maintaining the packed rectilinear shape. Our system will always generate packed rectilinear layouts, at the expense of adherence to the emphasis schema. The last two layout rows show the same phrases as Figure 1.

height, or vertically by scaling to the same width and stacking. For horizontal alignment, place the layouts with the words that come later in the phrase to the right of those with words that come earlier. For vertical alignment, place the layouts with the words that come later in the phrase below those with words that come earlier. Optical margin alignment and other spacing adjustments can also be made in this step.

3. **Recording Emphasis Metrics:** Create a characteristic vector for each layout that contains the value of a given metric (for example, height) for each word in the phrase.

4. **Prioritisation:** Sort the layouts according to their emphasis adherence scores.

5. **Presentation:** Present the layouts with the best emphasis adherence scores to the user.

## 3.1 Tree Construction

Our technique for generating all possible packed rectilinear layouts of a phrase uses a tree structure similar to the image layout algorithm, BRIC [4]. The key difference in our algorithm is the presence of additional geometric and layout constraints that are inherent in typographic layouts. Typographic layouts need to be designed with constraints on reading order and there is less flexibility in aspect ratio for each of the elements.

Each layout variation can be characterized by a tree and its alignments, where each leaf node represents a word and each internal node represents a vertical or horizontal alignment between its subtrees. For each subdivision of the phrase into non-empty subsequences, we recursively compute all possible subdivisions for each of the children. Unlike BRIC [4], our generated trees are not exclusively binary trees. This allows us to create layouts where more than two words are packed sequentially at the same width or height.

### 3.1.1 Big Schröder Numbers

The constraints imposed by word aspect ratios and reading order allow us to predict exactly how many variations of each phrase are possible. The number of possible layouts for a phrase consisting of $n$ words follows the sequence of Big Schröder numbers [3]. The first ten terms of the Big Schröder number sequence are 1, 2, 6, 22, 90, 394, 1806, 8558, 41586, 206098, which is an exponential sequence. Big Schröder numbers describe the number of ways a rectangle can be divided into $n + 1$ rectangles using $n$ distinct guillotine cuts, which mirrors how packed rectilinear layouts are essentially subdivisions of a rectangular layout outline [18].

## 3.2 Layout Variation Generation

All layouts can be constructed by alternating between vertical and horizontal alignments on each level of a tree structure. Nodes that share a common parent have the same height in the case of a horizontal alignment and width in the case of a vertical alignment. Figure 2 shows an example where the word TYPE is placed in a horizontal configuration with a subtree containing a vertical arrangement of the rest of the words in the phrase.

For each subdivision of the phrase into non-empty subsequences, we recursively compute all layouts for each of the subsequences. Starting from the leaf nodes, we generate layout variations of the whole phrase by placing a layout from each of the subsequence sets horizontally or vertically adjacent to one another. In horizontal alignments, words that are later in the phrase go to the right of the other words and in vertical alignments, words that come later in the phrase are stacked below earlier words. If there are $t$ tree structures generated, then there are $2t$ possible layouts because of the vertical and horizontal layout options. The alignments alternate between all vertical and all horizontal in a given level because a tree where a parent and its children have the same alignment is equivalent to one where the children have been moved to be siblings of the parent.

### 3.2.1 Word Order

In the tree construction process, we determine the order of the placement of the children using the order of the words with which they are associated: the recursive layout for a later subsequence of words must be to the right of the layout for the earlier subsequence, or below it. The resulting layout always has words that are later in the phrase placed to the right of, or under, preceding words. This follows the reading order convention for text in English, which is a Z-shaped reading order left-to-right, top-to-bottom.

### 3.2.2 Spacing

Leading is the baseline-to baseline vertical distance between lines of text. It is often specified as a fraction of the text size, which makes it difficult to determine leading when a display text layout uses multiple font sizes. We allow the user to define spacing for leading and horizontal space between words, and have an option to use the default horizontal spacing for the given font in the case of consecutive words that are the same height.

Optical margin alignment, or margin kerning, is the process of adjusting the horizontal spacing of a letter that overhangs on the margin of a piece of text to create the appearance of being aligned flush with the edge [22]. In packed rectangular display text layouts, this optical alignment is necessary for each word to achieve an optically aligned packing. We created a table of horizontal offsets, similar to an optical margin kerning table, to indicate the offsets required so that the edge of the word appears flush with the edge of the overall layout.

## 3.3 Layout Prioritization

With the large number of layouts generated, we can prioritize certain layouts and display a subset. We prioritize layouts based on the Euclidean distance between $\vec{e}$ and layout attributes $\vec{c}$. $\vec{e}$ is a vector of $n$ numbers representing the relative emphasis of each word in the phrase. The numbers are positive and do not need to be unique. $\vec{c}$ represents the values of any parameterized attribute, or characteristic, of the words in the phrase. We focus on word height in the examples presented in this work, but other attributes such as font weight and colour could also be used. We normalize $\vec{e}$ and $\vec{c}$ prior to comparing the euclidean distance between them.

Given an emphasis schema $\vec{e}$ and characteristic vector $\vec{c}$, both of size $n$, the Euclidean distance can be calculated:

$$d(e,c) = \sqrt{\sum_{i=1}^{n}(c_i - e_i)^2}$$

Note $d(e,c)$ can also be calculated using other distance metrics such as cosine similarity, but we did not find that using them made a noticeable difference in layout quality empirically. The final emphasis score, $E$, can be calculated as a linear combination of the values of $d(e,c)$ for all the characteristic vectors that the designer wishes to include. Users can specify the number of layouts they would like to see, and the algorithm will select that many matches with the smallest value of $E$.

## 3.4 Implementation

Our algorithm was implemented as a design tool in Processing[1] using the Geomerative library.[2] It is currently calibrated for the Verdana Bold font, but can be adapted for other fonts. We chose Verdana because Josephson et al. [12] found that Verdana was the most readable among their selection of fonts, and it is recommended for displaying letters and digits with high legibility [6]. Layouts can be exported as or .png or .svg files to support vector-based design in other software tools.

---

[1] https://processing.org/
[2] http://www.ricardmarxer.com/geomerative/

While our algorithm is exponential in the number of words, it works well with display text, which generally has fewer than ten words. Figure 3 shows a series of example layouts generated using our tool with varying emphasis schemas. The number of possible layouts increases quickly with each additional word, but this is unlikely to be a computational issue for display text layouts with ten words or fewer. Although exact performance depends on many factors, in one execution of the implementation of the algorithm on a consumer-grade 2.50 GHz processor, our tool took 66 milliseconds to generate all layouts and select the 5 layouts that best fit the emphasis schema for 4 words (22 variations). On longer phrases of 9 words (41,586 variations), the runtime was 82.5 seconds.

## 4 INTERVIEW STUDY

We conducted semi-structured interviews with five design experts to better understand design practices and preferences in packed rectilinear layouts, and to validate the efficacy of layouts generated by our tool. The goals of these interviews were:

- to understand designer preferences for packed rectilinear layouts
- to develop a hierarchy of visual emphasis methods
- to evaluate the efficacy of our layout prioritization method

This design study was split into two sessions: a within-subjects experiment involving web-based design tasks and a semi-structured interview to clarify the responses from the experiment comments. The web-based task primed the designers to think about designing packed rectilinear layouts before the interviews.

### 4.1 Participants

We recruited participants (3 female, 2 male) using the Adobe Illustrator Prerelease Forum, and selected participants with at least 10 years of professional design experience, with an average of 22.4 years of design experience. Participants received $100 CAD for successful completion of the study.

- P1 is a teacher with over 30 years of experience with graphic design and typesetting.
- P2 is an illustrator and multidisciplinary designer with over 10 years of graphic design and typesetting experience.
- P3 is a designer with 21 years of experience in graphic design and 15 years of typesetting experience.
- P4 is a Workflows and Adobe Instructor with 25 years of graphic design and typesetting experience.
- P5 is a graphic designer with 26 years of graphic design and typesetting experience.

### 4.2 Procedure

This user study was divided into a priming task followed by a semi-structured interview and design task with an experimenter. The web-based task was hosted on a Google Firebase server and created using the JsPsych[3] framework in JavaScript.

#### 4.2.1 Priming Tasks (20 Minutes)

First, the participant completed a *Scaling Task*. In this web-based activity, they were asked to scale a word relative to another word using a slider until they were at certain scales relative to one another (Figure 4). The interface had no visual guidance tools presented on-screen. There were three different target scales (0.5×, 2×, and 3×) across four different words of varying lengths (NO, CATS, EAT, and GRASS) for a total of 12 scaling tasks per participant. This task was designed to test which metric, such as height, width, or area, designers used to determine relative size and the degree to which it matched their actual selections.

Second, the participant completed a *Ranking Task*. They were given an emphasis schema, and asked to rank five layout designs

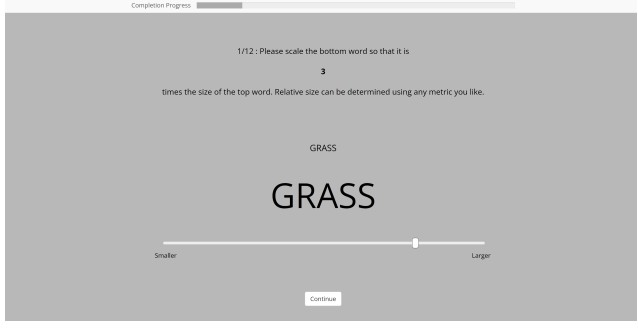

Figure 4: The scaling task with a scaling factor of 3.

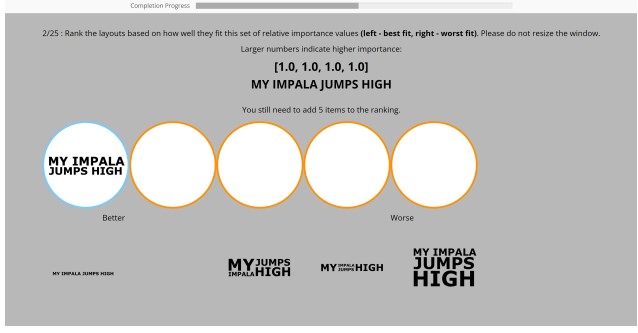

Figure 5: The ranking task with an emphasis schema of $(1, 1, 1, 1)$ and the phrase "MY IMPALA JUMPS HIGH". The circles represent rankings from left (highest) to right (lowest).

for the same phrase from best to worst according to how they fit the emphasis schema. This was done by dragging the image of the layout into an ordering (Figure 5). Participants were also asked about how much they liked the first and second choices in their ranking and could provide further explanation through a free-response box.

For the ranking task, we used five 4-word phrases, "ALL FROGS GO HERE", "ALL HORSES LOVE GRASS", "MY IMPALA JUMPS HIGH", "NO CATS EAT ORCAS", and "SOME CATS LIKE DOGS", each with five different emphasis schemas, $(1, 1, 1, 1)$, $(1, 1, 2, 1)$, $(1, 2, 3, 4)$, $(4, 3, 2, 1)$, and $(3, 2, 5, 1)$. These phrases have a variety of word-length distributions; for example, all words are the same length in "SOME CATS LIKE DOGS". We decided to limit the user study to phrases with four words to avoid having the phrase length interfere with the participants' judgement of the results between trials. Each participant completed 25 ranking tasks covering all combinations of phrase and emphasis schemas. For each task, we selected the top five layout variations based on scores from our tool and presented them to the participant in randomized order.

The order of the ranking task was grouped by emphasis schema. After each group of five phrases with the same emphasis schema, participants were asked to describe the strategies they used to rank the designs. The experimenter later used these responses to guide the semi-structured interview. This task was meant to evaluate the scoring function of the tool and how closely it matched designer expectations.

#### 4.2.2 Design Interview (40 Minutes)

The semi-structured interviews focused on six main themes related to the design of packed rectilinear layouts:

- *Readability*: What makes a layout readable? What are the considerations that must be made to ensure designs are understandable at different scales?

---

[3]https://www.jspsych.org/

- *Ambiguity*: Which factors cause layouts to have ambiguous reading order or meaning?
- *Alignment*: How should words of different scales be aligned in packed rectilinear layouts?
- *Spacing*: What determines leading and spacing between words when there are words of various sizes in a layout?
- *Emphasis*: Which factors can be used to emphasize certain words in a layout?
- *Scaling*: How is relative scaling between words determined?

Designers were encouraged to share their computer screens and create designs to illustrate the ideas that they discussed in these design sessions. For example, the experimenter prompted some of the designers to resolve reading order ambiguity in a given design and present their version of the layout. Examples generated by designers are discussed below.

## 5 RESULTS

Because of a technical error, the data for P2 was not recorded. However, their comments are included in the discussion below.

### 5.1 Scaling Preferences

We found that designers used height to judge the relative size of different words. The meaning of "size" in the scaling task prompt was intentionally ambiguous so that designers would use the size metrics that conformed to their internalized rules for text layout. Some possible correlates to emphasis include word height, area, length, and diagonal length. As seen in Figure 6, the user selections aligned better with estimation based on height than estimation based on area, which was calculated using $width \times height$.

The average error between user selection and height determined by the scaling value of the given task was -17.29% ($\sigma$=18.45, one outlier at 0.5x scale removed). We also asked the designers which strategies they used to determine relative scaling and all participants responded that they used the height of the word to determine size.

P1 judged relative scale by finding a tall letter with a flat top such as T to use as a benchmark. For words with no such letter available such as WOW or COO, they reported that they squinted and looked at the word upside down to see the perceived edges of the word without being distracted by its meaning or familiar shape. P2 also reported using a similar technique to exclude the overshoots of the rounded letters from their analysis of the shape.

All of the design experts that we interviewed mentioned "eyeballing it", or optical compensation, in reference to the spacing between two words of different font sizes and with the scaling tasks. P2 expressed how they used the overall height of the letter as a baseline in their mind to compare font scaling, but the designers did not follow references as strictly as we had previously imagined.

### 5.2 Ranking Preferences

We compared the preferences of designers in the layout ranking portion of the study with the emphasis adherence of our tool using Spearman's rank correlation coefficient, $\rho$.[4] Comparing the rankings given by designers and the rankings determined by our tool gave $\rho = 0.99$, which indicates a very high level of agreement between the designers and our proposed rankings. This suggests that our scoring algorithm is able to identify adherence to emphasis schemas.

---

[4]$\rho$ ranges between -1 for low correlation between two rankings and 1 for high correlation between rankings.

### 5.3 Semi-structured Interview

#### 5.3.1 Readability

Across all the design experts that we interviewed, the consensus was that readability was the most important consideration for display text. The designers considered left-to-right as the dominant direction that readers' eyes will move, followed by top-to-bottom. P3 explained that readers "naturally just read left [to] right, at least in Western language." P2 also expressed similar ideas about the default reading order for readers of English. For P1, absolute scale played a key part in readability, and by extension, how they ranked their preference for a layout. If any of the words in the given examples were too small to comfortably read, they automatically ranked it lower than the other layouts. To ensure readability at different scales for display text layouts, P3 talked about how they would shrink the canvas to simulate reading the layout from very far away.

#### 5.3.2 Ambiguity

Reading order ambiguities arise when there are deviations from the usual Z-shaped reading order that most readers of western languages are accustomed to seeing, which prioritizes left-to-right and then top-to-bottom reading. Deviations from this reading order without additional ordering cues can confuse the reader and negatively affect understanding. When asked to elaborate on their preferences for reading order, P5 said "I'm never going to read down, I'm always going to read across unless there's a break, or some other visual clue that those things go together like color, or different font."

The designers had several approaches for reducing ambiguity in layouts. One option for reducing ambiguity is to group words based on a certain attribute. During the free design portion of the interview, P4 created a layout that led users to read down by grouping based on different fonts and weight (Figure 7b) Another technique is to increase spacing between groups to make them distinct visual elements.

#### 5.3.3 Alignment

In packed rectilinear layouts, all words on the borders of the layout must be aligned to create a straight edge. Through interviews with designers, we found that this is usually determined using some form of optical margin alignment, with or without the use of the alignments defined by the font. P5 discussed how they often relied on optical bounds instead of inking boundaries to determine alignment. For example, if a word began with the letter "O" and was on the left edge, they were inclined to move it slightly more to the left to let the curve hang over the edge of the layout.

P2 introduced an interesting example of putting the vertical axis of the layout on a slant (Figure 7a). While our tool does not currently support these layouts, a tilt factor could be added to the algorithm for specific fonts that do not have a perfectly vertical y axis.

#### 5.3.4 Spacing

Leading and spacing are usually font attributes, but these are often manipulated by designers in display text layouts. These attributes are often designed with body text in mind so they are often irrelevant for display text. The spacing between words is highly dependent on the specific design, so there was less consensus between designers. In general, our participants used leading and spacing that were the same height and width, and used the default spacing for one of the fonts as a size reference.

P1 reported that their method for determining the approximate spacing between words in a layout with varying font sizes and packing is to take the standard space between words in the smallest font and use that as the size for leading and horizontal spaces. P2 had a slightly different approach of using double the default leading between the smallest words in the layout.

In order to separate two groups of words, P2 said that the space between groups should be "at least the length or the width of one of

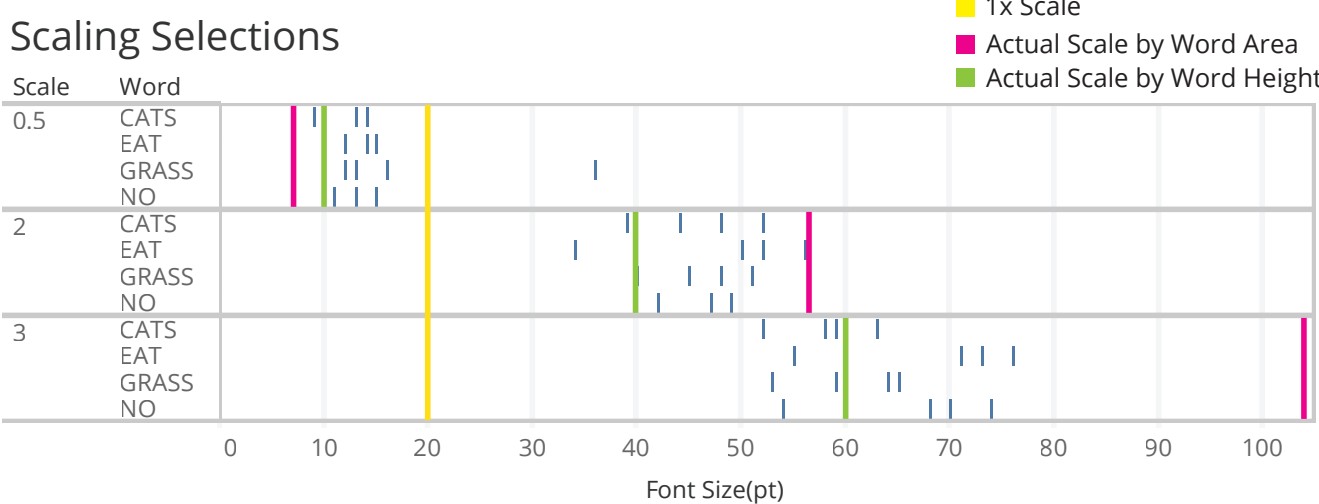

Figure 6: Scaling task selections for all participants, grouped by intended scale. The base scale was 20pt.

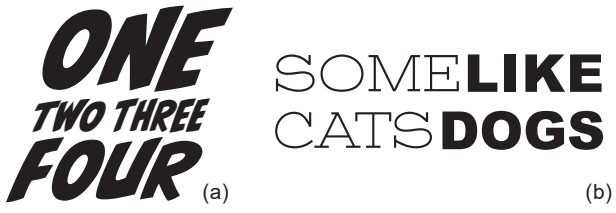

Figure 7: (a) An example of a layout that has a slanted vertical axis (b) An example of a layout that says SOME CATS LIKE DOGS which could be mistakenly read as SOME LIKE CATS DOGS. Grouping through font choice reduces ambiguity.

the largest characters." They also expressed how leading could also be doubled to create a vertical separation between word pairs.

P4 mentioned how font weighting also affects the amount of space they choose to add between words, "when words are bolder the designer tends to give the word more space to let it breathe."

### 5.3.5 Emphasis

Emphasis relies on the contrast between a word and its surroundings. It can be achieved through changing many different attributes such as the font, weight, colour, size, and placement. Size is the emphasis technique that we focused on in the priming task, but the designers in our study provided suggestions on how they use other techniques, depending on design needs. When asked about in-situ emphasis techniques that would not alter the layout, P1 offered insights on a rough hierarchy of emphasis techniques. Their top preference was adjusting the word weight, followed by editing the font of the emphasized word, and lastly adjusting colour. When asked about factors that affect emphasis in a layout, P3 said "I think scale is probably more important than placement."

## 6 DISCUSSION AND FUTURE WORK

### 6.1 Text Attributes

In this work, we focused on using word height as a proxy for emphasis. Other factors, such as colour or contrast differences between words, font weight, or using italics, can also affect the level of emphasis on a given word in a phrase. We only evaluated variations in word heights due to the exponentially increasing number of possible layout variations for each attribute, but future work might explore how these attributes can be used in conjunction with height to create varying levels of emphasis for a given word. For example, the saturation of the colour of each word can be used as another characteristic vector in addition to size-based metrics. The prioritization step would take the linear combination of all the characteristic vectors before comparing the result to the emphasis schema.

### 6.2 Variable Fonts

In the freestyle design portion of the interviews, many of the designers chose to use variable fonts to change the horizontal span and aspect ratio of words. Variable fonts, or OpenType Font Variations [2], are fonts with continuously adjustable parameters. While our tool did not take advantage of variable font weighting, it is a promising direction of future exploration. Variable fonts allow designers to change the emphasis of a certain word in a layout without changing the relative positions of each word in the layout but this would require parameterizations to create better emphasis metrics to determine $\vec{c}$.

During the free design portion of the study, P1 used variable fonts to make fine-grain adjustments to word weighting. In particular, they increased the weights of words that had a smaller font size to give all words in the layout similar weight despite differing sizes.

### 6.3 Rotation

Rotation was suggested by P1 and P4 as a way to de-emphasize certain words. Our packing algorithm could also be used for text rotated 90 degrees because the underlying principle of using the aspect ratio remains the same. However, we did not investigate the effect of rotational variations using our tool because it would have drastically increased the number of layouts that we considered. A pilot study for this project found that rotation slows reading speed and can be used to de-emphasize words in a layout. Future work might explore how rotations affect designers' preferences for layouts and how to model the resulting visual relevance.

### 6.4 Semantics

In this work, we discussed emphasis of words without a direct connection to semantics. With real-world design tasks, there is often a connection between semantic importance and emphasis. Language models could be used to detect the most important words in a layouts automatically and provide a starting point for users to specify their

emphasis preferences. For example, articles such as "the" or "an" are unlikely to require emphasis in a layout. Semantics could also affect the placement of words as different clauses of the phrase might require separation. In future enhancements, semantic breaks could be entered into the algorithm to create wider gaps between different clauses and reduce reading order ambiguity.

## 6.5 Optimization and Tree Pruning

The current implementation of the algorithm iterates over all possibilities of the layout, but performance could be improved by caching repeated subtrees during the tree generation process to avoid repeated computation. For example, if a short word of lower importance is stacked on top of a longer word with higher importance, we know that all of the subtrees containing this combination will have the wrong relative emphasis between those two words. The trees containing that subtree can therefore be excluded from the layout generation process.

Another alternative to achieve faster results would be to randomly generate a subset of possible layouts as opposed to enumerating through the exponentially increasing number of possible layouts. The algorithm could then explore this subset of layouts, find the layouts that match the emphasis schema the most closely, and then iterate on those results.

## 6.6 Ambiguity and Filtering

Ambiguity filters could be used to discard certain layout variations prior to the ranking and prioritization step. The primary source of reading order ambiguity is where non-consecutive words in the phrase have a similar height and are approximately aligned horizontally. Designers might wish to enable other filters, such as a minimum word height or maximum size disparity between words. The creation of a layout variation and its derivatives can be abandoned early if any of these characteristics are found during the layout process.

## 6.7 Fine Tuning

While our tool provides a starting point for designers, much of design is in details that are specific to each design project. The current implementation of our tool does not provide users with dynamic control over parameters such as the space between words. Future implementations of the tool might provide further support for designers to tune the generation of variations to reduce the space of variations to only include layouts that better align with their vision.

The layouts produced by our tool would likely be post-processed by designers in professional design programs to further reduce ambiguity and enhance aesthetic qualities. Our tool currently outputs editable vector files, but this algorithm could be implemented as a plugin for professional design tools such as Adobe Illustrator to create a seamless design experience in one application.

## 7 CONCLUSION

In this work, we present a new tool for automatically generating packed rectilinear display text layouts and prioritizing layout variations based on emphasis schemes. We also distill insights on creative typesetting from a set of interviews with expert designers. The automatic generation and prioritization of design variations allow a designer to explore all combinations of packed rectilinear layouts for a given phrase without the need for manual alignment and resizing. Automatic typographical layout tools introduce designers to possible layouts that would have otherwise been too time-consuming to explore. These suggestions can serve as a starting point for designers when creating packed rectilinear display text layouts.

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
