# OpenReview forum: "Generating Packed Rectilinear Display Text Layouts with Weighted Word Emphasis"
_graphicsinterface.org/Graphics_Interface/2023/Conference_SD — GI 2023 - second deadline_

### Official Review · Reviewer_so6Q · 2023-04-23
**review for rectilinear display text layouts**

**Rating:** 5
**Confidence:** 4

**Review:**

Paper Summary.
The paper introduces a method for generating rectilinear text layouts. In order to enable various layout designs, the proposed method takes use of a phrase tree structure inspired from an existing image layout algorithm. During this layout generation process, reading order and spacing are considered. The produced layouts are further prioritized according to the Euclidean distance between the user assigned emphasis weights and layout attributes (e.g., text height). Two qualitative experiments have been conducted to validate the usefulness of the method, including a scaling task and a ranking test. A semi-structured interview study has been conducted to analyze the users’ preferences in readability, ambiguity, spacing, emphasis and scaling.

Paper Strengths.
1.	A new solution is proposed to generate rectilinear text layouts, which provides a good starting point to improve the efficiency of text layout design.
2.	A priority metric is proposed to evaluate the quality of the generated layouts.
3.	The interview study claims the importance of reading orders and spacing for text layout design.
4.     The results are nice.

Paper Weaknesses.
The new solution is quite simple and ad-hoc. Some deeper analysis could be discussed, e.g., the relationship between the text scale (length of the texts) and the computation complexity of the solution. It would be better if the word ordering and spacing can be systematically formulated as constraints into the text layout generation process.

Some important references are missing.
1.	Balancing Font Sizes for Flexibility in Automated Document Layout
2.	Spatial Text Visualization Using Automatic Typographic Maps
3.	Review of Automatic Document Formatting
4.	ShapeWordle: Tailoring Wordles using Shape-aware Archimedean Spirals
5.	Content-aware Generative Modeling of Graphic Design Layouts
6.	Interactive By-example Design of Artistic Packing Layouts

Since the method is simple, the paper is easy to follow in general. But I think it can be improved in the following aspects.
1.	Some sections have only a short paragraph, which should be combined together.
2.	Pseudo-code or framework diagram would be helpful to explain the method.
3.	Some typo errors: “a emphasis”, “-17.29%”.

---

### Official Review · Reviewer_mS3A · 2023-04-23
**adequate submission, could be expanded**

**Rating:** 6
**Confidence:** 4

**Review:**

This paper proposes an approach to rectilinear text layout. Given a rectangle and a short phrase with importance weights attached to individual words, its algorithm subdivides the rectangle into subrectangles where each word can be placed. All combinations of subdivisions are considered and scored, and the top-scoring layouts can be shown to a designer. Interviews with graphic designers, primed by a task evaluating sample layouts generated by the paper's tool, provided recommendations and considerations for font layout.

Overall I mildly liked the paper. It poses a carefully-constrained design problem and produces a solution, judged by graphic designers as reasonable. The paper's text is reasonably organized and the exposition of the algorithm is clear. My main concern is that the contribution may be rather slight, owing to overconstraining the initial problem and then not thoroughly investigating the problem as posed; see below.

I gave the paper a marginally positive rating, which captures my assessment reasonably well. I hope that (if the paper is accepted) the authors can use the revision period to address at least some of my comments below. I do think the paper's tidy treatment of its constrained problem puts it above the bar for GI.

That said, I will spend most of the review describing opportunities for improvement. I see four areas where the paper could be improved; these are roughly arranged into importance order below.

1. More thoroughly testing the system

I am disappointed that so many examples used only four words. My uninformed observation is that tools like Adobe's Magic Layout are often used for much longer phrases and sentences, on the order of 5-12 words. The motivating example (Figure 1) has 7. Showing behaviour for more combinatorially challenging cases would go a long way towards demonstrating the value of the technique for regular users.

Having introduced a problem ("How to build a healthy/decent relationship with food") in the introduction, the paper does not revisit it. That is a significant missed opportunity to demonstrate the effectiveness of the technique -- or (less likely but I suppose possible) points to a major gap in the system, if it cannot adequately resolve this case.

Using such small layouts limited the value of the user study. It would have been good to test the system more generally rather than sticking to four-word phrases. Longer phrases produce more challenging layout problems, better justifying computer assistance.

2. Better integrating the interviews into the paper

The paper seems to be divided into two parts, only weakly related: the description of the algorithm for layout generation, and then the designer interviews. I first thought that the interviews would involve an evaluation of the computer-generated layouts, and evaluation is listed as a goal (bullet at the start of section 4), but the descriptions of the interviews did not lead to much evaluation as far as I can tell; the interview themes (4.2.2) do not lend themselves to such an evaluation. I guess the evaluation is meant to point to section 5.2? This can be better signposted, if so, and in any case is not an evaluation of the layouts per se, but of the scoring system (i.e., layouts that the tool would not have generated were not considered). The study is somewhat redundant because the unstated assumption is that the scoring system works: only the top 5 rated layouts were considered, and if a better layout was scored at #10, say, that would not have been discovered.

The authors have distilled some design advice and shown some examples of alternative layouts; this is great and could inform future work, but seems rather disconnected from the algorithm described in the first half of the paper.

3. Justifying the design problem

The rectangle subdivision approach makes sense given the design problem that 1) there is a fixed space to use, that 2) must be exactly filled by the text.  I am not very sure about either assumption. In many cases, the initial box size and aspect ratio can be adapted to the text, at least to some degree; letting the text poke out of the box (for letters like O or V, where there is little weight at the right and left) is possible. Similarly, even if the text cannot leave the box, there can be a margin of blank space around it, which in some cases can lead to a better layout.  Alternative information channels (changing color, changing font between words) are not considered. This makes some sense as a way of simplifying the computation, but these alternatives are indeed considered by designers, as the interviews indicated.

I do not expect the authors to revise their method to address these considerations, but rather to discuss the design constraints they imposed, how they relate to ecologically-valid design tasks, and to acknowledge the resulting limitations of the approach.

4. Improving the technique for layout decisions

The system exhaustively searches all possible layouts, with exponentially increasing cost as the phrase length increases. It seems as if many possible layouts are going to have low scores, though, and that it should be possible to terminate some tree expansions early with a branch-and-bound style strategy. In general, there are many alteratives to exhaustive search (including both pruning and randomized strategies) and some discussion and perhaps implementation of these could be worthwhile, perhaps in a followup paper if the authors continue this line of work. Exponential scaling is a disaster.

---

### Official Review · Reviewer_nX73 · 2023-04-24
**Nice paper but lacks technical depth**

**Rating:** 5
**Confidence:** 3

**Review:**


This paper describes a technique for automatically generating layouts for text. The layouts provide emphasis on important words using a weighting vector. The authors recruit professional designers to evaluate and guide their algorithm design. The criteria and survey for these interviews with professionals appear to be well-chosen.

This is a well-written, well-motivated paper that includes some interesting insights from designers, such as the predominant importance of height for size judgements and left-to-right/top-down word ordering for English writing.

Although I enjoyed this paper, many of the ideas could be better explained and demonstrated. Specifically:

1. In section 2, the authors mention that techniques such as O'Donovan et al. are computationally expensive. Given that this technique is based on exhaustive generation and ranking, what are the running times?

2. In section 2 and 3, the authors mention that BRIC also uses a tree structure to manage vertical and horizontal layouts. However, BRIC is based on a binary tree but this work does not. What is the rational behind using (or not using) a binary tree?

3. How would colors be combined with height to provide emphasis? This paper would be much stronger if more text attributes could be considered by the algorithm.

4. The authors give a good example from Adobe in Section 1 that shows why generating good layouts with emphasis as challenging. But the paper does not show how their own technique would perform on the same example.

5. In section 5.1, could you define what is meant by "area". Is it width *height of the font? Or is it pixel coverage?

---

### Official Review · Reviewer_hHNQ · 2023-04-24
**OK study, underwhelming algorithm**

**Rating:** 6
**Confidence:** 4

**Review:**

The paper proposes a system to generate a text layout with given emphasis per word, filling a box with some leading and spacing. They generate all possible combinations of layouts, filter the ones not satisfying the word order (and other) constraints, and then pick the best one with the minimum Euclidean distance between their emphasis vector and font size (or width or height). They validate their system via a user study.

I agree that it would be great to have this as a tool. The proposed algorithm is very naïve and completely unscalable (if I understand correctly, laying out 10 words or more is pretty much impossible due to exponential complexity), but perhaps for the motivational-quote--kind of texts the length of 9 words would be enough? However, to me the value of this paper is not as much in the algorithm, as in the user study. I find the scaling study rather clever and the results (Fig.6) interesting -- perhaps this can be looked further into in the follow-up works. The interview discussion of spacing is also quite interesting. It's really more of an HCI paper rather than graphics in as sense.

In general, I'm not quite impressed by the system or the algorithm, but I think there is some value in the study.

Detailed comments:
- Sec. 3.0.1: at this point it is unclear how Big Schroder numbers are relevant (There is no detailed discussion of constraints up to this point)
- The actual algorithm is never described anywhere, I had to guess from all the details given in various places. Please add a pseudocode even if it's trivial?
- Measuring Euclidean distance between something relative (emphasis vector) and something absolute (font size) makes no sense really because the scales are very different. I guess in this case it worked out because 1,2,3,4 are close enough to typical font sizes, but this might explain the artifacts we see (e.g., in Fig. 3 for 1,2,3,4 the leftmost figure clearly does not have "goal" larger than "the", which should contradict the emphasis).
- (minor) Please do not start sentences with math (beginning of Sec 3.2)